# Emerging Functions of lncRNA Loci beyond the Transcript Itself

**DOI:** 10.3390/ijms23116258

**Published:** 2022-06-02

**Authors:** Hober Nelson Núñez-Martínez, Félix Recillas-Targa

**Affiliations:** Instituto de Fisiología Celular, Departamento de Genética Molecular, Universidad Nacional Autónoma de México, Ciudad de México C.P. 04510, Mexico; hnunez@ifc.unam.mx

**Keywords:** long noncoding RNA, gene expression, chromatin, CRISPR-Cas9

## Abstract

Thousands of long noncoding RNAs (lncRNAs) are actively transcribed in mammalian genomes. This class of RNAs has important regulatory functions in a broad range of cellular processes and diseases. Numerous lncRNAs have been demonstrated to mediate gene regulation through RNA-based mechanisms. Simultaneously, non-functional lncRNA transcripts derived from the activity of lncRNA loci have been identified, which underpin the notion that a considerable fraction of lncRNA loci exert regulatory functions through mechanisms associated with the production or the activity of lncRNA loci beyond the synthesized transcripts. We particularly distinguish two main RNA-independent components associated with regulatory effects; the act of transcription and the activity of DNA regulatory elements. We describe the experimental approaches to distinguish and understand the functional mechanisms derived from lncRNA loci. These scenarios reveal emerging mechanisms important to understanding the lncRNA implications in genome biology.

## 1. Introduction

The development of high-throughput sequencing technologies has shed light on transcriptome knowledge leading to the identification of thousands of transcriptionally active genomic regions that do not derive from protein-coding genes but produce long noncoding RNAs (lncRNAs) [1,2,3]. In general, lncRNAs share much of the molecular mRNA characteristics; they are transcribed by RNA polymerase II (RNAPII), spliced, and polyadenylated. Nevertheless, lncRNAs are less abundant and less stable as well as less conserved than mRNAs with a preferential nuclear localization [1,4]. The regulatory mechanisms of lncRNA transcripts in genome regulation have been extensively reviewed [5,6]. For instance, lncRNA-protein interactions modulate the accessibility or binding of interacting proteins to their target sites in the genome in a broad range of biological processes and diseases [7,8]. This mechanism relies on RNA-based functions where the lncRNA transcripts carry out the regulatory function. 

Typical regulatory DNA elements such as promoters and enhancers drive lncRNA transcription or are embedded into lncRNA loci [9,10]. Surprisingly, these regulatory elements are, in general, more evolutionary conserved than their lncRNA transcripts [11,12]. This observation raises the question of whether many lncRNAs transcribed from regulatory elements are functional transcripts or non-functional byproducts of lncRNA loci activity. Examples illustrate these observations and underpin the notion that many lncRNA loci produce non-functional lncRNA transcripts [13,14]. However, the process of transcription or the activity that originated from their associated regulatory elements may contribute to local gene regulation.

In this review, we discuss the underlying mechanistic principles associated with the activity of lncRNA loci beyond the RNA-dependent function. In particular, we discuss recent insights into the regulation originated from the activity of transcription and the regulatory elements of hosted lncRNA loci. We also provide an updated overview of the available methods for interrogating the mechanistic scenarios derived from these observations to distinguish the functional components of given lncRNA loci.

## 2. *Cis*-Acting lncRNAs Regulate Local Gene Expression

In general, lncRNAs are coexpressed with adjacent or neighboring protein-coding genes [1,15]. Although this co-expression is frequent between any pair of chromosomal neighbors, which suggest that it is not dependent on the coding or noncoding nature of the genes. This observation is just a correlation analysis; it does not suggest or refute if the lncRNA transcript is functional. In some cases, lncRNA transcripts exert regulation on the vicinity where they are transcribed (a *cis*-acting mechanism) [16,17]. *Cis*-acting lncRNAs regulate gene expression at the local level by remaining tethered to their transcription site to modulate gene regulation through chromatin looping, modulate the binding of transcription factors, and the establishment of histone modifications to their target regions [17,18,19,20]. Additional to lncRNA transcripts, the local regulation of gene expression by lncRNA can be mediated by processes associated with their synthesis. These regulatory processes do not necessarily depend on the lncRNA. Here, we describe these processes in more detail.

## 3. Gene Regulation by the Act of lncRNA Transcription

Gene regulation may be influenced by neighboring transcription, where the act of transcription of a given locus per se can regulate nearby genes [21]. This neighboring transcription, therefore, can affect gene expression positively or negatively through two main mechanisms: (i) Nucleosome remodeling; or (ii) modulating the deposition of histone marks, binding of transcription factors, and RNA polymerase II complex [22]. Importantly, these mechanisms are not mutually exclusive. Strikingly, the process of lncRNA transcription is similar to mRNAs, which raises the question of how lncRNA transcription affects the genes located nearby [23]. Below we describe how the lncRNA transcription influences local gene regulation.

### 3.1. LncRNA Transcription Instructs Nucleosome Displacement and Gene Silencing

Noncoding RNA transcription can be originated not only from intragenic but also from intergenic genomic regions [2]. This observation suggests that noncoding transcription may promote chromatin remodeling that can spread toward the vicinity of functional regulatory DNA elements [24]. As a consequence, RNAPII passage increases nucleosome density downstream from the transcribed regions towards nucleosome-depleted regions such as enhancer and promoter elements. The increased nucleosome density compacts chromatin, which physically prevents the access of transcription factors and RNAPII machinery to their targets [25] (Figure 1A). This suppressive effect of noncoding transcription is known as transcriptional interference, which has been extensively described [25,26]. In *S. pombe*, the phosphate-responsive permease gene (*tgp1+*) is regulated by the transcription factor Pho7 in response to phosphate stimulus. The lncRNA *nc-tqp1* is transcribed upstream of *tgp1+* upon phosphate starvation. The activity of noncoding transcription originated from *nc-tqp1* inducing *tgp1+* silencing by an increase in nucleosome occupation over the promoter region, which physically prevents Pho7 binding [27]. Similar to *nc-tqp1*, noncoding transcription originated from serine-induced lncRNA *SRG1* silences *SER3* by increasing nucleosome density over *SER3* promoter in response to serine stimulus. The increased nucleosome compactness interferes with the binding of transcription factors on the *SER3* promoter [28]. Thus, *nc-tqp1* and *SRG1* are upstream lncRNAs whose transcription act interferes with the binding of transcription factors by modulating nucleosome displacement over promoter regions.

### 3.2. Chromatin Landscape Is Influenced by lncRNA Transcription

In addition to the gene silencing induced by the physical displacement of nucleosomes, lncRNA transcription may influence the chromatin state through modifying the profile and deposition of histone marks, binding of TF, and the activity of RNAPII. During the progression of transcription, the RNAPII associates with chromatin remodeling complexes, transcription factors, and histone chaperones, contributing to the chromatin state dynamics during transcription [29]. There are several scenarios where overlapping lncRNA transcription influences gene transcription through local chromatin modifications (Figure 1B). 

In mammals, the *Tcra* locus is involved in V(D)J recombination during thymocyte development [30]. This locus produces a noncoding transcription that encompasses the promoters of proximal and distal Jα-Cα gene segments. As a consequence, noncoding transcription facilitates the activation of proximal Jα-Cα gene segments through the deposition of active histone modifications (H3ac, H3K4me2, H3K4me3, and H3K36me3) and chromatin accessibility. In contrast, the distal Jα-Cα segments are repressed [31]. Intriguingly, this noncoding transcription demarcates chromatin states (active and inactive) necessary for the recombination and diversification of T-cell receptors (TCRs). 

In *D. melanogaster*, the Polycomb Response Elements (PRE) silence genes by targeting Polycomb Group (PcG) complexes, which are necessary for homeotic gene regulation [32]. Noncoding transcription overlaps with the *Fab-7* PRE in the bithorax locus preventing the repressive effect by the PcG complex [33]. Thus, lncRNA transcription modulates the access of chromatin remodeling complexes to the chromatin. The chicken lysozyme gene (*LYZ*) is transcribed under lipopolysaccharide (LPS) stimulation [34]. The insulator protein CTCF binds to an upstream enhancer element blocking the activation of *LYZ* by the enhancer. Upon LPS stimulation, the upstream enhancer becomes active and transcribed into the lncRNA LINoCR leading to CTCF eviction from the enhancer and subsequent *LYZ* activation [35]. Oppositely, the cardiac-enriched lncRNA Upperhand (*Uph*) is transcribed upstream to *Hand2*. The *Uph* transcription is necessary to maintain the binding of GATA-4 and the activation of two overlapped enhancer elements-driven *Hand2* gene expression [13]. In such cases, lncRNA transcription modulates the binding of TF and the activation of hosted regulatory elements. These mechanisms illustrate a more complicated picture where even if the act of transcription originating from lncRNA loci is associated with activation of its hosted lncRNA, it does not discriminate among the activation or repression outcomes.

### 3.3. Emerging Functions of lncRNA Transcription and Chromatin Folding

Although architectural proteins such as CTCF and cohesin complex mediates chromatin structuration through facilitating a loop formation between the enhancers and promoters [36], lncRNA transcription can influence local chromatin topology. The act of transcription can induce CTCF and cohesin displacement from chromatin loop anchors leading to disruption of chromatin interactions. As a result, local chromatin undergoes a loss of compaction and switching of inactive to active compartments [37]. For instance, during T-cell lineage development, the lncRNA *ThymoD* is transcribed from an upstream intergenic region from the *Bcl11b* gene, which harbors an enhancer element. The act of transcription repositions the enhancer element from the nuclear periphery to the nuclear interior, triggering the enhancer activation. Moreover, *ThymoD* transcription facilitates CTCF/Cohesin complex occupancy over the enhancer and the *Bcl11b* promoter, which juxtaposes enhancer–promoter communication into a loop structure [38] (Figure 1C). Accordingly, lncRNA transcription may occur in a three-dimensional chromatin context necessary to activate and bring closer regulatory DNA elements to proximal genes through shape chromatin structure. It is important to consider lncRNA transcription as a regulatory hub where the production, transcripts, and regulatory elements can contribute to regulating the chromatin landscape as well as gene regulation. In summary, the molecular mechanisms of how noncoding transcription shapes chromatin folding remain to be unraveled. A further characterization of examples supporting an interplay between lncRNA transcription and genome folding could help to understand how the act of transcription shapes genome folding and maintain nuclear architecture.

## 4. LncRNA-Embedded Regulatory Elements Drive Local Gene Expression

Enhancers are functional DNA elements involved in gene activation independent of their distance or orientation to their target genes through a mechanism known as chromatin looping [39,40,41]. Importantly, these genomic regions are targeted by transcription factors involved in gene activation and repression, which turns out on specific spatiotemporal gene expression on their target genes. A significant proportion of lncRNAs are transcribed from enhancer regions [10,42], which indeed explains their tissue-specific expression pattern as a product of the activity of their associated regulatory elements [32]. It is important to consider that the presence of regulatory DNA elements such as promoters and enhancers within lncRNA loci can regulate local gene expression independently of the act of transcription or sequence of the transcribed lncRNA (Figure 2A). In particular, the erythroid-enriched lncRNA *Lockd* is localized downstream of the *Cdkn1b* gene locus. Lockd is transcribed from a regulatory element with apparent promoter-associated histone marks, which physically interacts with the *Cdkn1b* promoter. The act of transcription and *Lockd* transcript are dispensable; however, the regulatory element embedded within *Lockd* locus appears to act as an enhancer that regulates *Cdk1b* [14]. Likewise, the presence of regulatory elements within lncRNA loci can function as a boundary to limit the regulatory effect of distal regulatory elements with proximal genes. The lncRNA *PVT1* promoter prevents the interaction between *PVT1* intragenic enhancers with the *MYC* promoter by attenuating *MYC* overexpression (Figure 2B) [33]. Importantly, a considerable fraction of lncRNA genes appears to influence local gene expression through the enhancer-like activity of lncRNA promoters, which have also been observed in mRNAs [43]. The *Bendr* lncRNA promoter acts as a distal regulatory element necessary to activate *Bend4* gene expression. Even though the *Bendr* promoter would not be classified as an enhancer, it acts through a similar mechanism. Notably, gene promoters with dual functions as enhancers have been described before [43,44]. It remains to elucidate how gene promoters acquire enhancer-like functions and if they are predominantly encoded into lncRNA loci. In summary, these examples underpin how regulatory DNA elements such as enhancers and promoters within lncRNA loci contribute to local gene regulation.

## 5. Interrogating Functions of lncRNA Loci

The characterization of lncRNA loci is challenging due to their multifunctional roles in gene regulation, as we illustrated above. Some characteristics of lncRNA transcripts such as abundance, cellular localization, sequence composition, or conservation inaccurately predict their biological functions. 

Thus, many studies support the fact that some lncRNA transcripts are functional molecules rather than by-products of transcription. So far, computational and experimental efforts have been committed to characterizing the multiple mechanisms of actions that can come out from lncRNA loci demonstrating that lncRNA loci are clearly complex through either the RNA transcript itself, the act of transcription, or the activity of regulatory DNA elements encoded in the locus. On the other hand, a series of exciting genetic tools have been developed to distinguish the role of lncRNA transcripts from their transcriptional process or the activity of its associated regulatory DNA sequences, which in turn can determine the essential elements from a given lncRNA locus required for its regulatory activity and even more its biological relevance [45]. Thus far, logical experimental approaches have been proposed to distinguish the diverse molecular basis of a given lncRNA locus (Table 1), thus far, many of them have emerged from the study of protein-coding genes.

Additionally, we describe experimental strategies and the methodologies used to dissect the possible lncRNA-associated mechanisms of a particular locus (Table 1). Also, we highlighted bona fide examples of lncRNA loci representing those mechanisms and the experimental strategies used to unravel their molecular basis. Importantly, in some cases the experimental approaches proposed in this section do not exclude the possibility that a particular locus has more than one mechanism of action.

### 5.1. RNA-Based Mechanism

To dissociate the function of a given lncRNA transcript from possible regulatory elements encoded in the DNA locus that are required for its cis-regulatory activity, the habitually used approach is to knock down the RNA molecules. The RNAi system has been the most used strategy to directly target the transcript of lncRNA by inducing RNA degradation and subsequent loss of function of lncRNA transcripts [60]. More generally, nuclear localization of many lncRNAs and their low abundance make them less efficient to be depleted compared to protein-coding genes [61]. Also, similar approaches have been adopted using chemical modified oligomers such as LNA (locked nucleic acid), PNA (protein-nucleic acids), and ASO (antisense oligonucleotides) [62,63,64], which appear to be efficient to knock down nuclear-abundant lncRNAs. However, that efficiency depends in part on the strategy to deliver into the cells and if the knockdown effect can be recapitulated through the cell passages or whole organisms. Recent technologies based on genome editing have emerged and been adopted to study lncRNA function [65]. The most common, the CRISPR-Cas9 system has been adapted to interrogate the role of lncRNA loci by deletion of whole locus or specific DNA sequences such as promoters, exons, or splicing sites [3,13]. Further, the CRISPR-Cas9 system coupled with homologous end repair (HDR) has been adjusted to replace or insert sequences into lncRNA loci, such as exons, polyadenylation, or degradation signals. These elegant tools pave the way for obtaining a molecular basis for dissecting the function of lncRNA transcripts. However, it is important to take a careful look at any lncRNA locus as a potential hospitable DNA region of annotated and unannotated regulatory DNA elements. Accordingly, experimental strategies include the removal or replacement entire lncRNA locus can affect not only the production of the underlying lncRNA transcript but also the activity of putative regulatory DNA elements and the act of transcription, which in turn causes misunderstanding of regulatory mechanisms or cellular phenotypes that do not only depend on the lncRNA transcript [3]. Crucially, it is necessary to explore the existence of regulatory elements before choosing the regions to be edited. Hence, experimental approaches focused on affecting the levels of the lncRNA transcript but do not perturb the DNA sequence like RNA interference or degradation, the CRISPR-Cas13 system, or exogenous overexpression are recommendable. Note that exogenous overexpression is applicable only for trans-acting but not cis-acting lncRNAs due to cis-acting lncRNAs accumulate locally in proximity to their target genes.

If a cis-effect is observed upon deletion or replacement of the entire lncRNA locus but not by knocking down the RNA transcript, then the locus represents two regulatory scenarios: it encodes cis-acting DNA elements or the transcriptional activity are required to exert its function, whereas the RNA transcript is often dispensable. In our experience, initial experimental strategies including deletion of promoter and exon sequences by CRISPR-Cas9 are the most reliable, efficient, and rapid approaches to dissecting possible regulatory scenarios of a particular lncRNA locus.

### 5.2. DNA Regulatory Elements within lncRNA Loci

Regulatory DNA elements (promoters, enhancers, and transcription binding sites) embedded into lncRNA loci can exert cis-regulation separately from the lncRNA transcript or the act of transcription [12,52]. Promoter deletion o replacement are initial strategies for lncRNA ablation, however these procedures led to changes in lncRNA expression levels as well as the act of transcription, and even cis effect can be caused by enhancer activity of the deleted promoter [12]. An important consideration when using promoter or whole locus deletion is looking into local cis effects at the same locus. To further investigate if the underlying promoter or enhancer within the lncRNA locus exerts a regulatory effect, a catalytically dead version of Cas9 (dCas9) fused to domains that activate (CRISPRa) or inactivate (CRISPRi) lncRNA transcription has been initially used to manipulate lncRNA levels, however this approach can affect the activity of interrogated regulatory elements by remodeling the chromatin landscape but not the DNA sequence [53,66]. Moreover, alteration of regulatory elements by the aforementioned approaches may cause local promoter-enhancer rewiring such as promoter competition between lncRNA loci with neighbor genes [54]. Certainly, annotated lncRNAs are transcribed from their associated promoter sequences, which does not rule out the possibility that many lncRNA loci have cis-effects on gene regulation where the promoter sequences are involved. Thus, local gene regulation by lncRNA loci may be due to the regulatory influence coming from the promoter or enhancer sequences. Of note, combining experimental approaches focused on RNA or DNA-based mechanisms are required to determine the molecular basis behind lncRNA loci.

In parallel, examining available high throughput sequencing data such as chromatin accessibility (DNase I and ATAC-seq), histone modifications (H3K4me3, H3K4me1, H3K27ac), and TF binding from ChIP-seq can help to clarify the context in which a lncRNA locus can exert its function. Moreover, chromosome conformation capture (Hi-C, capture Hi-C, Hi-ChIP, or ChIA-PET) data can inspect to elucidate the possible interacting pattern of the putative regulatory element with potential target genes [67].

### 5.3. The Act of Transcription

The process of transcription by itself can drive local gene expression regardless of the synthesized transcript as described above. To unravel whether the transcription process coming from a locus of interest exerts cis-regulation, transcription termination is the frequently used method. Inserting polyA-terminator sequences immediately from the lncRNA TSS allows premature transcription termination [12,14]. Using this strategy, different lengths of transcription can be generated to address the effect of transcription on regulatory elements or neighboring genes [12,68,69]. However, in some cases, not all the inserted terminators are efficient to abolish downstream transcription [12]. Whether it is a consequence of location-dependent polyA insertions remains to be addressed. Biallelic polyA insertion or the use of multiple tandem polyA sites can enhance transcription termination [69]. Alternatively, the catalytically dead Cas9 (dCas9) positioned along with the lncRNA TSS or gene body has been used to block transcription and inhibit RNA synthesis without perturbation of DNA sequence [54,66,70], which in effect is the most reliable method to study the effect of transcription without disturb DNA sequences of regulatory elements. Even though both methods inhibit lncRNA transcription, the synthesis of the lncRNA transcript is attenuated. Subsequent experimental approaches such as RNAi should be applied to distinguish the regulatory effect of the act of transcription or the RNA transcript.

Collectively, all the experimental approaches described before should be applied to unraveling the complex molecular mechanism of a lncRNA locus, which in turn led us to identify if the candidate lncRNA locus is, in fact, a functional RNA transcript. However, it could be experimentally complex, especially in investigating the dynamic regulation of lncRNA loci during cell differentiation systems or organism development. Finally, these elegant and carefully experimental approaches can lead us to dissection of all the possible regulatory scenarios coming from lncRNA loci. Examples of lncRNA locus where one or more mechanistic principles may act together to regulate gene expression in a particular context have been reported [3,49,59].

## 6. Concluding Remarks and Future Perspectives

Our knowledge about gene regulation has evolved with the discovery of the lncRNA genes. Even though evidence demonstrates that some lncRNA transcripts are byproducts of transcription or transcriptional noise, others are functional RNA molecules involved in gene regulation. Even more, the DNA sequences where lncRNA emerges and the process of transcription can influence local gene expression. This underlines the importance of applying genetic tools to study the RNA and DNA as regulatory modalities emerging from lncRNA loci, in which several possible modes of action can arise from disentangling the molecular basis of lncRNA loci. The examples described here highlight the need to apply rigorous experimental approaches to identify functional lncRNA transcripts as well as putative regulatory DNA elements associated with gene regulation. These emerging mechanisms of action by themselves can exert gene regulation with relevant biological functions such as cell differentiation, imprinting, development, and diseases. Fortunately, emerging exciting genome editing tools like CRISPR-Cas9 coupled together with information about the epigenetic landscape have enabled distinguish the lncRNA mechanisms of action. Although these efforts, it remains to clarify the molecular mechanisms and the functional roles of lncRNA which are complex loci given the existence of processes associated with their production and the activity of regulatory DNA elements that originated it. Future research will allow characterizing on genome-wide scale potential non-functional lncRNA transcripts as well as new putative functional regulatory mechanisms associated with lncRNA loci. It will be also interesting to unveil if these regulatory mechanisms observed in lncRNA loci also take place in other noncoding loci such as miRNAs, piRNAs, or cirRNAs and their contribution to cell biology and disorders.

## Figures and Tables

**Figure 1 ijms-23-06258-f001:**
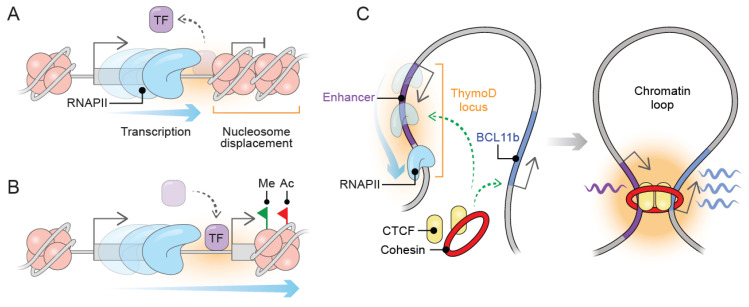
Transcription from lncRNA loci remodels the chromatin landscape to induce gene regulation. (**A**) The act of transcription from lncRNA loci can induce gene silencing transcriptional interference, which leads to increasing nucleosome density downstream from transcribed lncRNA loci to physically prevent TF binding and avoid the access of RNAPII machinery. (**B**) Chromatin landscape can be reprogrammed by lncRNA transcription. Overlapped transcription drives the deposition of histone modifications and dynamic TF binding. (**C**) Local chromatin folding can be restructured through the dynamic binding of structural proteins. Transcription from *ThymoD* locus overlaps an enhancer element, which leads to CTCF/Cohesin binding to the enhancer and the downstream BCL11b promoter, allowing promoter-enhancer interaction throughout a chromatin looping structure. Transcription factors (TF), RNA polymerase II (RNAPII), histone methylation (Me), and histone acetylation (Ac).

**Figure 2 ijms-23-06258-f002:**
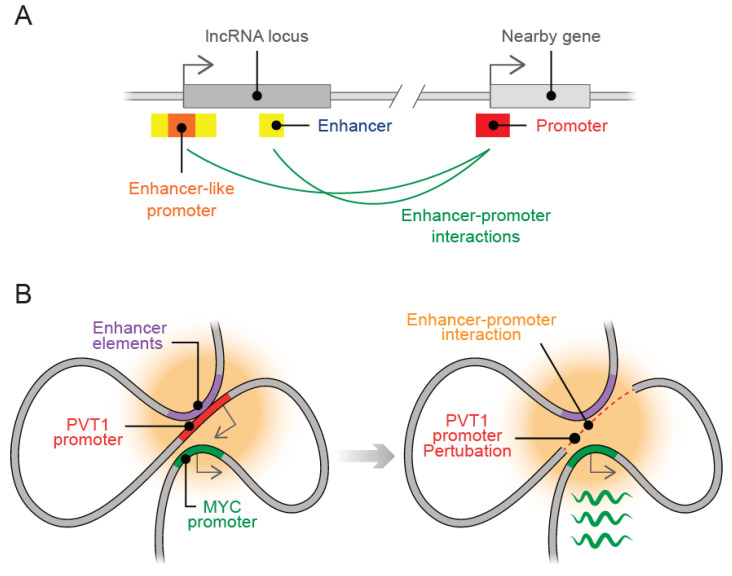
Regulatory elements within lncRNA loci influence local gene regulation. (**A**) Promoters with enhancer-like activity and enhancer elements embedded in a particular lncRNA locus can induce a regulatory effect at distant genes. In some cases, neither the act of transcription nor the RNA transcript is necessary to distal gene regulation by the regulatory element. (**B**) LncRNA promoters can function as boundary elements to limit gene regulation by distal regulatory elements such as enhancers. *PVT1* promoter prevents enhancer-promoter communication between intragenic enhancers within *PVT1* lncRNA and *MYC* promoter. *PVT1* promoter inactivation or mutation enables enhancer-promoter interactions and subsequent *MYC* overexpression.

**Table 1 ijms-23-06258-t001:** Experimental methods to for discriminating the regulatory mechanisms of lncRNA loci.

Experimental Strategy	Methods	lncRNA-Associated Mechanism			Reference
		RNA-Based Mechanism	The Act of Transcription	The Activity of Embedded RE	
Entire locus deletion	CRISPR-Cas9	Yes	Yes	Yes	[3,14,46]
Promoter deletion or replacement	CRISPR-Cas9 and HDR	Yes	Yes	Yes	[12,36,47,48]
Deletion or replacement of exon sequences	CRISPR-Cas9 and HDR	Yes	No	No	[13,48,49]
Deletion of splicing sites	CRISPR-Cas9	Yes	No	No	[12,50,51]
Deletion of RE	CRISPR-Cas9	No	No	Yes	[52]
Activation or inactivation of RE	CRISPRa and CRISPRi	Yes	Yes	Yes	[53,54,55]
Transcription termination	PAS insertion and CRISPR-dCas9	Yes	Yes	No	[12,14,36]
RNA interference or degradation	RNAi, CRISPR-Cas13, ASO, PNA and TWI	Yes	No	No	[33,56,57,58,59]
Exogenous overexpression	Plasmid expressing lncRNA transcript	Yes	No	No	[59]

HDR (homologous direct repair), RE (Regulatory element), RNAi (RNA interference), ASO (antisense oligonucleotide), PNA (peptide nucleic acid), TWI (twister ribozyme).

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
