# Peer review of "Emerging Functions of lncRNA Loci beyond the Transcript Itself"

_ijms, 2022, doi:10.3390/ijms23116258_

Round 1

Reviewer 1 Report

This is a nice, well written and condensed review on an important topic in gene regulation. The authors have made a great affort to present updated and relevent information.

Although I must admit that every author has its own preferences regarding the examples to include in the manuscript, I have missed a description of the very first "regulatory ncRNAs" detected, the called "sterile transcripts"  that originate from genes of the immune system (such as the heavy chain-Ig) and were though to act as transcriptional/reorganization drivers of these genes. What is the current view of these "sterile transcripts"?.

Furthermore, the manuscript would greatly benefit of the inclusion of a few Figures/drawnings to facilitate understanding of this dense topic.

Author Response

Responses to the referees (round 1)

Reviewer 1: Although I must admit that every author has its preferences regarding the examples to include in the manuscript, I have missed a description of the very first "regulatory ncRNAs" detected, the called "sterile transcripts" that originate from genes of the immune system (such as the heavy chain-Ig) and were thought to act as transcriptional/reorganization drivers of these genes. What is the current view of these "sterile transcripts"?

We appreciate the reviewer’s observations and we agree about we missed the description of the early evidence of noncoding transcription in gene regulation. During B cell development and immune response, B cells undergo class switching to rearrangement of gene segments for the constant region of the immunoglobulin heavy chain and properly production of immunoglobulin diversity [1]. This process is accompanied by switching transcription initiating upstream of the switch regions (positioned 5’ from the CH genes) and terminates at the corresponding CH gene. It turns out the production of processed transcripts, also known as sterile transcripts [2]. This class of transcripts does not encode functional immunoglobulins nor do their exon sequences appear to be necessary, however sterile transcription is required to establish a permissive chromatin environment for the class switching [3]. In line with this evidence, early sterile transcription termination prevents the proper exchange of gene segments and class switching [4]. A contradictory finding suggested that the processing (splicing) of sterile transcripts is required for class switching but not the process of transcription [1, 5]. A similar observation supports the linking between lncRNA splicing and local gene regulation [6]. Such contradictory findings described above may be due to the use of cell models and experimental strategies. We do not discard that both mechanisms orchestrate, even the sterile transcripts could function as regulatory units of class switching during the production of immunoglobulins. Importantly, sterile transcription is controlled by its regulatory elements (promoter or enhancers) supporting the idea that it is a regulatory mechanism that needs to be properly controlled during immune response [3]. It is not yet clear how sterile transcription and sterile transcript exert their regulatory role, so It will require further studies.

In the section “3.2 Chromatin landscape is influenced by lncRNA transcription” of our review, we described a similar example about the regulatory role of noncoding transcription on Tcra locus and its role in recombination and diversification of T-cell receptors (TCRs). However, we do not refer to the products of noncoding transcription as “sterile transcripts” nor described their presumable role in immunoglobulin production as the authors of this paper did not describe in the original paper. Even though, they pinpoint the role of noncoding transcription but not the function of noncoding RNA transcripts.

References

  1. Hein, K., et al., Processing of switch transcripts is required for targeting of antibody class switch recombination. J Exp Med, 1998. 188(12): p. 2369-74.
  2. Lennon, G.G. and R.P. Perry, C mu-containing transcripts initiate heterogeneously within the IgH enhancer region and contain a novel 5'-nontranslatable exon. Nature, 1985. 318(6045): p. 475-8.
  3. Su, L.K. and T. Kadesch, The immunoglobulin heavy-chain enhancer functions as the promoter for I mu sterile transcription. Mol Cell Biol, 1990. 10(6): p. 2619-24.
  4. Abarrategui, I. and M.S. Krangel, Noncoding transcription controls downstream promoters to regulate T-cell receptor alpha recombination. EMBO J, 2007. 26(20): p. 4380-90.
  5. Harriman, G.R., et al., IgA class switch in I alpha exon-deficient mice. Role of germline transcription in class switch recombination. J Clin Invest, 1996. 97(2): p. 477-85.
  6. Engreitz, J.M., N. Ollikainen, and M. Guttman, Long non-coding RNAs: spatial amplifiers that control nuclear structure and gene expression. Nat Rev Mol Cell Biol, 2016. 17(12): p. 756-770.

Reviewer 2 Report

This review is on an important topic but suffers from a lack of clarity in some parts which needs to be addressed. The English need to checked thoroughly since some of the sentences do not make sense.  It lacks depth in certain sections. The rauthors claim to  address experimental approaches to distinguish and understand the functional mechanisms derived from lncRNA loci. A table summary of these methods would really helpful addition to the manuscript. The diagrams are nicely done.

Author Response

Responses to the referees (round 1)

Reviewer 2: This review is on an important topic but suffers from a lack of clarity in some parts which need to be addressed. The English need to be checked thoroughly since some of the sentences do not make sense.  It lacks depth in certain sections. The authors claim to address experimental approaches to distinguish and understand the functional mechanisms derived from lncRNA loci. A table summary of these methods would really helpful addition to the manuscript. The diagrams are nicely done.

We appreciate the reviewer’s observations. We agree to carefully check the sentences, and clarify and depth the information in certain sections. We assess those observations in the manuscript. So, we can visualize by turning on “track changes” in Microsoft Word. Concerning experimental approaches, in “Table I. Experimental methods to discriminating the regulatory mechanisms of lncRNA loci” we summarize several available exciting experimental approaches to dissect the lncRNA-derivated mechanisms. By the way, we also provide references regarding the experimental strategies to dissect the lncRNA-associated mechanisms.

Reviewer 3 Report

The authors briefly review the functional mechanisms derived from the regulation of lncRNA loci on chromatin remodeling and update potential experimental approaches. The topic is very interesting, particularly in the use of CRISPR systems, and the manuscript is well structured and easy to follow. However, I think the authors should add more details about the proposed methods (too short and general) and include their own experience. It would be interesting if the authors shared what strategy they are using in the lab. The authors can add additional material to expand concepts or even include current protocols. Include the limitations of the new strategies proposed and future trends.

Author Response

Reviewer 3: The authors briefly review the functional mechanisms derived from the regulation of lncRNA loci on chromatin remodeling and update potential experimental approaches. The topic is very interesting, particularly in the use of CRISPR systems, and the manuscript is well structured and easy to follow. However, I think the authors should add more details about the proposed methods (too short and general) and include their own experience. It would be interesting if the authors shared what strategy they are using in the lab. The authors can add additional material to expand concepts or even include current protocols. Include the limitations of the new strategies proposed and future trends.

We acknowledger the reviewer’s observations and suggestions. We incorporated more details and descriptions concerning the use of specific experimental approaches to unravel the multiple emerging mechanisms originating from lncRNA loci, which can visualize by turning on “track changes” in Microsoft Word. Moreover, we include the advantages and limitations of using the suggested experimental strategies. Furthermore, we highlighted the most reliable and recommended strategy for each underlying mechanism that is analyzed. Of note, we described our particular experience and recommendations for initial approaches to investigate the molecular basis and function of a particular lncRNA locus. Consequently we included additional references supporting such strategies, recommendations, and key concepts that can help us to disentangle the mechanisms of action of lncRNA loci. Finally, we added a section “6: Concluding remarks and future perspectives”, where we introduced these emerging mechanisms as mediators of biological processes and disorders.

Reviewer 4 Report

lncRNA transcripts are well-known regulators of gene expression in mammals and other “higher” organisms. These transcripts interact with both other RNA molecules and proteins. However, the act of lncRNA transcription by RNA polymerase II also influences the expression of nearby genes e.g., by modulation chromatin structure and occlusion polymerase and transcription factors. Drs. Núñez-Martínez and Recillas-Targa have submitted a brief review about the regulation originating from the “act of transcribing lncRNAs”. The review has the potential to inform mainly non-experts about this important type of regulation, but the current manuscript has several shortcomings.

·      Only 3 of the 38 references in the reference list are published in 2020 or later. A brief consultation with PubMed indicates that several other recent papers should be included to bring the review up-to-date.

·      The reference list is incomplete including only 38 references while the reference numbers go to 55.

·      The language needs a serious overhaul; many sentences being incomplete, non-interpretable, or grammatically incorrect. I realize that the authors are not native English speakers, so I hope that IJMS can provide some assistance. A non-inclusive list of examples are:

o   Page 3, line 4 from bottom: itself and the RNA-dependent?

o   Page 3, bottom line: we highlighted also provided an updated overview. Is something missing?

o   Page 4, lines 5-7: Although this co-expression is as frequent as between any pair of chromosomal neighbors suggesting that is not dependent on the coding or noncoding nature of the genes. Is something missing?

o   Page 4, line 3 from bottom: Both mechanisms are not mutually exclusive> These mechanisms …..

o   Page 5, lines 4-7: Noncoding RNA transcription can be originated from intragenic and intergenic genomic regions [2]. Therefore, noncoding transcription promote chromatin remodeling which can spread toward the vicinity of functional regulatory DNA elements> Noncoding RNA transcription can originate not only from intragenic, but also from intergenic regions [2], suggesting that noncoding transcription promote chromatin remodeling that can spread toward the vicinity of functional regulatory DNA elements.

·      Section 5 repeats several points already made in the previous sections

Author Response

Reviewer 4: lncRNA transcripts are well-known regulators of gene expression in mammals and other “higher” organisms. These transcripts interact with both other RNA molecules and proteins. However, the act of lncRNA transcription by RNA polymerase II also influences the expression of nearby genes e.g., by modulation chromatin structure and occlusion polymerase and transcription factors. Drs. Núñez-Martínez and Recillas-Targa have submitted a brief review about the regulation originating from the “act of transcribing lncRNAs”. The review has the potential to inform mainly non-experts about this important type of regulation, but the current manuscript has several shortcomings:

  1. Only 3 of the 38 references in the reference list are published in 2020 or later. A brief consultation with PubMed indicates that several other recent papers should be included to bring the review up-to-date.
  2. The reference list is incomplete including only 38 references while the reference numbers go to 55.
  3. The language needs a serious overhaul; many sentences being incomplete, non-interpretable, or grammatically incorrect. I realize that the authors are not native English speakers, so I hope that IJMS can provide some assistance. A non-inclusive list of examples are:

- Page 3, line 4 from bottom: itself and the RNA-dependent?

- Page 3, bottom line: we highlighted also provided an updated overview. Is something missing?

- Page 4, lines 5-7: Although this co-expression is as frequent as between any pair of chromosomal neighbors suggesting that is not dependent on the coding or noncoding nature of the genes. Is something missing?

- Page 4, line 3 from bottom: Both mechanisms are not mutually exclusive> These mechanisms …..

- Page 5, lines 4-7: Noncoding RNA transcription can be originated from intragenic and intergenic genomic regions [2]. Therefore, noncoding transcription promote chromatin remodeling which can spread toward the vicinity of functional regulatory DNA elements> Noncoding RNA transcription can originate not only from intragenic, but also from intergenic regions [2], suggesting that noncoding transcription promote chromatin remodeling that can spread toward the vicinity of functional regulatory DNA elements.

- Section 5 repeats several points already made in the previous sections

      We aknowledge the reviewer’s observations and suggestions. Concerning observations 1 and 2, We noticed that our summited manuscript has only 38 references, which do not coincide with our initial version of the manuscript. We guess that some references were missed during the first round of revision. Of note, and base on the reviewer suggestion we added recent references supporting examples of the mechanisms described in our review.

      Moreover, we realized that many sentences were incomplete, non-interpretable, or grammatically incorrect (concerning observation 3). Similarly, we noticed unexpected changes in many sentences that were not in our original manuscript. We corrected those observations in the manuscript. So, the editor can visualize by turning on “track changes” in Microsoft Word. According to these modifications, we consider that it is an improved version of our review based on these observations. Nevertheless, we do not discard any request for IJMS assistance after this round of revision.

      Finally, we also improve the description given in section 5. Which, in fact, includes repetitive information made in the previous sections.

Round 2

Reviewer 2 Report

the authors have done a good job with changes. I would suggest the add one more sentence on the importance of these molecules in disease and add a reference such as PMID: 32455975. 

Author Response

Reviewer 2: the authors have done a good job with changes. I would suggest the add one more sentence on the importance of these molecules in disease and adding a reference such as PMID: 32455975.

We aknowledge the reviewer’s observations and suggestions. We totally agree concerning the mentioned importance of lncRNA transcripts in disease and cell biology. In the introductory part of our review we added a sentence in which we mention:

The regulatory mechanisms of lncRNA transcripts in genome regulation have been extensively reviewed [5,6]. For instance, lncRNA-protein interactions modulate the accessibility or binding of interacting proteins to their target sites in the genome in a broad range of biological processes and diseases [7, 71]. This mechanism relies on RNA-based functions where the lncRNA transcripts carry out the regulatory function.”

Here, we pinpoint the importance of lncRNA transcripts as products of the transcriptional activity originating from lncRNA loci which have been substantially characterized and described in excellent recent reviews [1-6]. However, our review underlying the importance of the mechanisms derivated from lncRNA loci that exert gene regulation in an RNA-dependent manner. Additionally, we include references of examples of these mechanisms in diseases such as cardiac defects [7] and limb malformations [8]. We also mention the importance of these mechanisms as possible regulators of other types of non-coding RNA loci, such as circular RNAs.

References

  1. Ni, Y.Q., H. Xu, and Y.S. Liu, Roles of Long Non-coding RNAs in the Development of Aging-Related Neurodegenerative Diseases. Front Mol Neurosci, 2022. 15: p. 844193.
  2. Mabeta, P., R. Hull, and Z. Dlamini, LncRNAs and the Angiogenic Switch in Cancer: Clinical Significance and Therapeutic Opportunities. Genes (Basel), 2022. 13(1).
  3. Garcia-Padilla, C., et al., Molecular Mechanisms of lncRNAs in the Dependent Regulation of Cancer and Their Potential Therapeutic Use. Int J Mol Sci, 2022. 23(2).
  4. Beylerli, O., et al., Long noncoding RNAs as promising biomarkers in cancer. Noncoding RNA Res, 2022. 7(2): p. 66-70.
  5. Statello, L., et al., Gene regulation by long non-coding RNAs and its biological functions. Nat Rev Mol Cell Biol, 2021. 22(2): p. 96-118.
  6. Sun, J. and C. Wang, Long non-coding RNAs in cardiac hypertrophy. Heart Fail Rev, 2020. 25(6): p. 1037-1045.
  7. Anderson, K.M., et al., Transcription of the non-coding RNA upperhand controls Hand2 expression and heart development. Nature, 2016. 539(7629): p. 433-436.
  8. Allou, L., et al., Non-coding deletions identify Maenli lncRNA as a limb-specific En1 regulator. Nature, 2021. 592(7852): p. 93-98.

Reviewer 3 Report

I agree with the authors' comments and the paper has been improved.

Reviewer 4 Report

The revised version is much improved: Many up-to-date references have been added, and the language is largely corrected. The  manuscript is now fully understandable, but IJMS may want to do a minor check of the use of prepositions, articles, etc.